# Spirality: A Novel Way to Measure Spiral Arm Pitch Angle

Deanna Shields [1,*], Benjamin Boe [2], Casey Pfountz [3], Benjamin L. Davis [4], Matthew Hartley [5], Ryan Miller [6,7], Zac Slade [8], M. Shameer Abdeen [9], Daniel Kennefick [1] and Julia Kennefick [1]

1 Department of Physics & Arkansas Center for Planetary Sciences, University of Arkansas, 226 Physics Building, 835 West Dickson Street, Fayetteville, AR 72701, USA
2 Institute for Astronomy, University of Hawaii, 2680 Woodlawn Drive, Honolulu, HI 96822, USA
3 Department of Medicinal Chemistry, The University of Kansas, 1251 Wescoe Hall Drive, 4070 Malott Hall, Lawrence, KS 66045-7582, USA
4 Center for Astro, Particle and Planetary Physics (CAP³), New York University Abu Dhabi, Abu Dhabi 129188, United Arab Emirates
5 Physics Department, California Institute of Technology, 103-33, Pasadena, CA 91125, USA
6 NASA Headquarters, 300 E. Street SW, Washington, DC 20546, USA
7 Arlington Tech, Arlington Career Center, 816 S Walter Reed Dr, Arlington, VA 22204, USA
8 Free Geek of Arkansas, 521 W Ash Street, Fayetteville, AR 72703, USA
9 Department of Physics and Astronomy, Georgia Southern University, Statesboro, GA 30458, USA
* Correspondence: dshields@uark.edu

**Abstract:** We present the MATLAB code Spirality, a novel method for measuring spiral arm pitch angles by fitting galaxy images to spiral templates of known pitch. Computation time is typically on the order of 2 min per galaxy, assuming 8 GB of working memory. We tested the code using 117 synthetic spiral images with known pitches, varying both the spiral properties and the input parameters. The code yielded correct results for all synthetic spirals with galaxy-like properties. We also compared the code's results to two-dimensional Fast Fourier Transform (2DFFT) measurements for the sample of nearby galaxies defined by DMS PPak. Spirality's error bars overlapped 2DFFT's error bars for 26 of the 30 galaxies. The two methods' agreement correlates strongly with galaxy radius in pixels and also with *i*-band magnitude, but not with redshift, a result that is consistent with at least some galaxies' spiral structure being fully formed by $z = 1.2$, beyond which there are few galaxies in our sample. The Spirality code package also includes GenSpiral, which produces FITS images of synthetic spirals, and SpiralArmCount, which uses a one-dimensional Fast Fourier Transform to count the spiral arms of a galaxy after its pitch is determined. All code is freely available.

**Keywords:** data analysis methods; image analysis methods; galaxies; spiral galaxies; spiral arms; galaxy structure; galaxy evolution

## 1. Introduction

The global, best-fit pitch angles (*P*) of spiral galaxies are an ongoing topic of interest because they correlate to difficult-to-estimate properties such as bulge mass, central black hole mass, rotational velocity, and dark matter halo mass [1–4]. Pitch angle is easier to measure than, and can be used as an estimator for, more fundamental properties. Several methods exist of estimating pitch angles, and these methods roughly fall into a few categories: point sampling, one-dimensional Fast Fourier Transforms (1DFFT), two-dimensional Fast Fourier Transforms (2DFFT), and template fitting.

In the point sampling methods, individual points from spiral arms are fit to mathematical spirals, either by a direct least-squares fit [5] or by computing the tangent of the slope on a graph of $ln(r)$ vs. the azimuthal coordinate [6]. This has the advantage of using minimal, if any, computing power, but also discards potentially useful information by reducing the spiral arms to a set of far fewer points than the number of pixels in the image.

1DFFT [7] involves analyzing the azimuthal intensity on circles concentric with the deprojected galaxies. The phase angles of the symmetric 2-arm component are plotted vs.

ln(r), the pitch is extracted from the slope. Kendall et al. [8] subtracts out the axisymmetric components in order to enhance the spiral structure before performing 1DFFT on radial bins.

Saraiva Schroeder et al. [9] described a method of measuring a galaxy's spiral arm pitch angle using 2DFFT. The method measures both the strength and the pitch angle of the various modes (1-arm, 2-arm, etc.) between a given inner radius and a given outer radius. It should be noted that the mode actually describes rotational symmetry rather than the physical number of spiral arms. That is, the 2-arm mode measures the 180-degree symmetric component which usually, though not always, dominates in 2-arm spirals. The resulting pitch is the one that corresponds to the strongest mode, which hopefully matches the visual number of arms. The weaker modes are generally discarded.

2DFFT methods in general [1,9–12] decompose spiral arms into sums of logarithmic spirals of varying pitches. The number of spiral arms is taken to be strongest symmetry mode, which allows 3-arm spirals and 4-arm spirals to be analyzed in addition to grand design 2-arm spirals. However, for galaxies with more complex structures such as focculent galaxies, 2DFFT discards asymmetric features that may be of interest to the researcher.

Davis and Hayes [13] use a sophisticated computer vision algorithm to pick out spiral arm segments, and then models each segment as a logarithmic arc. Each segment may have its own segment length, pitch angle, etc.

In template fitting methods, spiral arms or arm segments are directly matched to spirals of known pitch. Puerari et al. [14] use template fitting to analyze individual arms or arm segments, which allows more detailed analysis of spiral structure.

Recently, Hewitt and Treuthardt [15] combine a parallelized 2DFFT code with galaxy tracings (potentially by citizen scientists) to create a hybrid manual/automated method.

### 1.1. Current Work

Seigar et al. [1] used 2DFFT to discover a correlation between a galaxy's spiral arm pitch angle and the mass of its central black hole. Davis et al. [10] noted that 2DFFT is luminosity-biased, so that the pitch angle near the inner radius is output. They therefore introduced a variable inner radius to the 2DFFT method. This method shows quantitatively how logarithmic the spiral arms are. It outputs the pitch of the most stable (i.e., the most logarithmic) radius segment. It also establishes error bars by considering the size of the stable radius segment relative to the galaxy radius, as well as the degree to which the stable segment is logarithmic. Berrier et al. [2] used the variable inner radius 2DFFT method to tighten the correlation between spiral arms and central black hole mass. The sample size was subsequently doubled and the black mass scaling relation refined by Davis et al. [3].

In this paper, we introduce Spirality, a MATLAB software which implements a novel algorithm for measuring spiral arm pitch angles that relies on best-fit matching to digital spiral templates. Unlike Davis and Hayes [13] and Puerari et al. [14], which analyze local segments of spiral arms, Spirality focuses on the global best-fit pitch.

The Spirality method does not rely on 2DFFT, so it does not force the user to choose a symmetry mode. This can be an advantage for galaxies that are not grand design. We adopt the variable inner radius method to test for logarithmicity and to establish error bars. We also introduce GenSpiral.m, a code for quickly generating FITS images of synthetic spirals with a wide range of properties. These spirals are useful for testing galaxy measurement and analysis codes.

Finally, we introduce SpiralArmCount.m, a code that uses a one-dimensional Fast Fourier Transform (FFT) to count a galaxy's spiral arms after the pitch angle is determined.

### 1.2. Motivation

A major aim in developing Spirality is for use in theory testing. The underlying physical causes of spiral arm structure in disk galaxies is still debated, and it seems appropriate to seek out ways in which measurement can decide between competing theories. Spirality is capable of quantifying two important aspects of spiral arm structure: It can measure

the pitch angle of the arms without assuming rotational symmetry, and it can count the number of arms.

The two main versions of density wave theory, the modal theory [16,17] and the swing amplification theory [18,19], suggest that young stars born in the spiral arms should move ahead of the density wave in the inner disk but fall behind in the outer disk, due to differential rotation. The result is that the star-forming dust (far infrared) and the short-lived massive stars (far ultraviolet) should be more loosely wound than the older, longer-lived stars (optical and near-infrared) [7]. The manifold theory, on the other hand, suggests that pitch angle should be independent of wavelength [20].

Building upon Pour-Imani et al. [21], Miller et al. [22] showed that for a sample of 16 galaxies, the pitch angle in the far ultraviolet (151 nm) is the same as the pitch angle in the far infrared (8.0 μm), and both of these pitches are systematically looser than the pitches in the optical and near-optical bands. This result strongly favors the modal density wave theory and the swing amplification theory over the manifold theory.

In order to discriminate between the modal density wave theory and the swing amplification theory, it is necessary to look more closely at their respective predictions. Berrier and Sellwood [23] and D'Onghia [24] observed that swing amplification demands that a denser disk cannot support multiple armed modes, with m > 3. In general, there should be a correlation between number of spiral arms and the density of the disk. Such claims can be investigated using SpiralArmCount, a part of Spirality package which can count a galaxy's arms in a simple and robust way.

Another theory-testing application involves the question of when spiral arms first formed. Elmegreen and Elmegreen [25] report that spiral structure began at around $z \sim 1.8$. Such high-redshift galaxies are often viewed as low-resolution images, so it is useful to have at least two independent methods of estimating pitch angle, such as 2DFFT and a template fitting method like Spirality.

Spirality has two main advantages. First, as will be shown, the pitch angle result describes the entirety of the galaxy in the measurement annulus, rather than stating only the dominant symmetry mode. Spirality also provides a clear way to count the spiral arms without assuming rotational symmetry.

## 2. Materials and Methods

A spiral's pitch angle $P$ is the angle between the spiral's tangent line and a concentric circle's tangent line (for an illustration, see Davis et al. [3], Figure 1). For a logarithmic spiral, $P$ is constant along the spiral arm, although for a physical spiral arm the pitch may vary along each arm and from arm to arm.

### 2.1. Spiral Coordinate System

A real orthogonal coordinate system with logarithmic spiral axes can be constructed. One set of axes has pitch angle $P$, where $-90° \leq P \leq 90°$. The orthogonal axes has pitch angle $P' = P \pm 90°$, where $-90° \leq P' \leq 90°$.

In order to construct one of the spiral coordinate axes, let a one-arm spiral be given by

$$r = e^{b\theta} \tag{1}$$

where $r$ is the radial polar coordinate, $\theta$ is the azimuthal polar coordinate, and $b$ is the so-called spiral constant.

The pitch angle, using the sign convention of astronomers, is then given by the spiral constant $b$ in Equation (1):

$$P = \tan^{-1}(-b) \tag{2}$$

The arclength S along the spiral from the origin to some outer radius $R$ is given by

$$S = \left| R\frac{\sqrt{1 + b^2}}{b} \right| \tag{3}$$

The spiral coordinate system then consists of many one-arm spirals, each given by

$$r = e^{b[\theta - \theta_0]} \tag{4}$$

Each axis has a unique phase angle $\theta_0$, such that $0 \le \theta_0 < 2\pi$.

The orthogonal set of axes is not computed for our purposes.

### 2.2. Computation Method

Our method finds a galaxy's best-fit pitch angle by matching the deprojected galaxy image [**input: FILE**] to a set of spiral coordinate systems (templates) with known pitch angles in degrees that vary from some user-defined lower value [**input: MINP**] to some upper value [**input: MAXP**] in discrete steps. The step size is defined by the user [**input: PSTEP**]. First, a single template is created.

The Cartesian pixel coordinates are recorded along each spiral axis. A set of evenly spaced points along the first spiral axis is established, with the point spacing in pixels determined by the user [**input: AxisPointSpacing**], for the purpose of recording pixel values. The axis should are spaced sufficiently close together so that a single pixel may be recorded several times. Then, the longer the path of the axis through a given pixel, the more times the pixel is counted. The mean pixel value along the axis is then recorded.

Figure 1 illustrates the process of finding the mean pixel value along a given spiral axis. The distance in pixels between the sampling points is called AxisPointSpacing, and is an input parameter that can be adjusted if computation time becomes an issue.

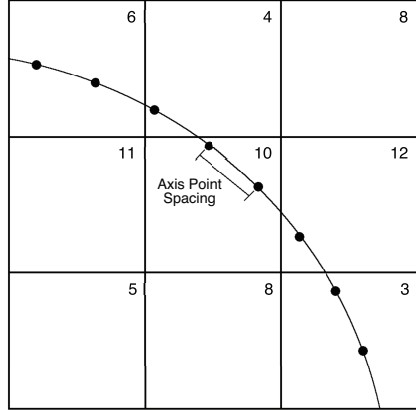

**Figure 1.** Illustration for computing the mean pixel value on a given spiral axis with a given pitch. The squares represent pixels on a galaxy image, and the numbers are the pixel values. The solid curve represents one spiral axis; the dots are the sampling points. The AxisPointSpacing input parameter is the distance, in pixels, between sampling points. The mean pixel value is the mean of all the pixels cut by the spiral, weighted by the number of sampling points in each pixel. For the spiral axis segment shown in this figure, the mean pixel value is $(2*6 + 4 + 2*10 + 12 + 2*3)/8 = 6.75$. The variance of these means, computed over all spiral axes for a given pitch angle, yields the fitting function for that pitch. When the axis pitch most closely matches the galaxy's pitch, the fitting function is maximized.

The process is repeated for the remaining axes. The number of axes is chosen by the user [**input: NAXIS**]; we recommend approximately $4\pi R$, where $R$ is the galaxy radius in pixels, so that the pixels at the outer edge of the measurement annulus get read, on average, twice. For particularly large galaxies, where computation time may become an issue, the number of axes can be reduced to $2\pi R$.

Once the mean pixel value for each spiral axis is recorded in the first template, the variance of toutput:hese means is computed. This is the fitting function. Its value is assigned to the pitch angle of the template. The process is then repeated for templates of

many different pitches, resulting in the fitting function (variance of means) vs. template pitch angle.

If the code is measuring a synthetic spiral or a particularly clean galaxy image, the fitting function shows a global maximum at the spiral's true pitch. However, the presence of even a single foreground star can produce a monotonically increasing background in the fitting function. The true pitch may therefore produce a local maximum, with the global maximum occurring at the edge of the graph. Visual inspection of the graphs is therefore recommended.

### 2.2.1. Error Bars

Error bars are determined in a manner similar to Davis et al. [10]. The user provides the pixel coordinates of the galaxy center [**inputs: X0, Y0**]. The pitch angle is measured on a series of annuli that are concentric with the galaxy. The galaxy's outer radius in pixels [**input: MSMT_OUTER**] is held constant, while the inner radius is varied from some lowest value [**input: MSMT_INNER1**] to some highest value [**input: MSMT_INNER2**] in discrete steps. The spacing in pixels between inner radii is given by the user [**input: InnerRadiusSpacing**]. The fitting function now spans two independent variables: template pitch angle and inner radius.

The code is run, and a region of inner radii with reasonably constant pitch is identified by the user [**output: PITCHvsINNER**]. The user then runs the code again, using only the stable range of inner radii. The average pitch is computed [**output: BESTFITPITCH**], and the standard deviation of the pitches in this region is computed. It is scaled by the ratio of size of the radius segment with stable pitch to the full radius segment of the spiral arms. The radius segment of the spiral arms is given by the user [**inputs: VIS_INNER, VIS_OUTER**]. The result is added in quadrature with the measurement precision [**input: PSTEP**], or the number of degrees between consecutive template pitch angles. The error bar is thus computed [**output: ERR**].

Because the error bar is larger if the logarithmic segment of the galaxy radius is smaller, the error bar is a first-order test of logarithmicity. If the user either overestimates or underestimates the length of the logarithmically stable radius segment, the error bar will suffer.

The measurement's precision, in degrees, is chosen by the user. We have found it useful to measure each galaxy twice. The first measurement is coarse (poor precision) but spans a wide domain of template pitch angles. The true pitch is visually estimated as a local maximum in the fitting function. Since the code outputs the global maximum, the output pitch may represent the edge of the graph rather than the true pitch. The galaxy is therefore remeasured with fine precision, spanning a narrow pitch angle domain in the region of the true pitch. On this domain, the true pitch is the global maximum, and is therefore output.

### 2.2.2. Measuring a Symmetric Component

For galaxies that are particularly noisy, are riddled with foreground stars, or for some other reason are difficult to measure, the code package includes SymPart.m, which returns the galaxy's 2-arm (180° rotational) component. Computation time for SymPart is negligible. The method is to pair each pixel with the pixel that is symmetric to it about the origin. The brighter of the pair is then reduced in value to the dimmer.

Measuring the pitch angle of the 2-arm component (mode) sometimes yields a more decisive answer than the galaxy as a whole. Moreover, if the image is not star subtracted, taking the symmetric component can be a quick way to eliminate most foreground stars. The disadvantage of taking the symmetric component is that it assumes 2-arm symmetry, while disregarding all information to the contrary.

SymPart can also yield the 3-arm component, by grouping each pixel with two other pixels at the same radial coordinate. The three grouped pixels form an equilateral triangle about the origin. The brightest two pixels in the group are reduced in value to the dimmest pixel. Using a similar method, the 4-arm component can also be taken.

In order to test SymPart, we added the resulting symmetric component to its residuals in hopes of recovering the original galaxy. The tests were successful for the 2-arm, 3-arm, and 4-arm components, but not for higher modes. We do not recommend using SymPart for modes higher than 4.

### 2.2.3. Measuring a Combination of Symmetric Components

For galaxies with only one or two foreground stars, the code package includes MultiSymPart.m, which quickly returns a combination of the 2-arm and 3-arm components. Computation time for MultiSymPart is negligible. The method is to compute the 2-arm component as described above, then compute the 3-arm component of the residual, and then add the two results. This can often eliminate a foreground star or two while disregarding only as much information as is necessary. The user can also include the 4-arm component.

As with SymPart, we tested MultiSymPart by adding the output to the final residuals in hopes of recovering the original galaxy. We do not recommend using MultiSymPart for modes higher than 4.

### 2.2.4. Counting the Spiral Arms

Once the pitch angle of a galaxy has been determined, the code package includes SpiralArmCount.m for counting the spiral arms. This code is useful for showing quantitatively whether or not a galaxy has an integer number of arms, particularly when a simple visual inspection of the galaxy gives an ambiguous answer. Computation time for SpiralArmCount is negligible.

SpiralArmCount requires the pitch angle as an input. The user can use either Spirality or any other method to determine the pitch angle.

The method is to analyze the image using a spiral coordinate system with the same pitch angle as the galaxy. The median pixel value $V_{med}$ is computed along each spiral axis. A graph of $V_{med}$ vs. coordinate axis phase angle $\theta_0$, where $0 \leq \theta_0 < 2\pi$, is available for the user's inspection.

The counting function is the FFT of $V_{med}$ vs. $\theta_0$. The resulting domain frequencies are converted to modes (1-arm, 2-arm, etc.), and the number of spiral arms is represented by the strongest mode.

Like 2DFFT, Spirality's SpiralArmCount function uses an FFT as its counting function. As with all FFT's, there are quirks.

First, the 1-arm mode analyzes a wavelength that spans the entire data set, which limits FFT's precision. The 1-arm mode for both Spirality and 2DFFT should therefore be eyed with caution.

Second, because SpiralArmCount relies on an FFT, it does not, strictly speaking, count the spiral arms. Rather, it outputs the rotational symmetry. Therefore, a three-arm galaxy with two bright, 180° symmetric arms and one dimmer, asymmetric arm will appear to both 2DFFT and to Spirality's SpiralArmCount function as a 2-arm galaxy because of its 2-arm symmetry. Likewise, SpiralArmCount would see a galaxy with three arms of equal brightness spaced 90° apart as a 4-arm spiral because of the 90° rotational symmetry.

The advantage of SpiralArmCount is that, in addition to the FFT, it also produces a graph of median pixel value vs. axis phase angle. On this graph, each spiral arm is represented by a local maximum. This gives the user a way to count the spiral arms independently of both the galaxy image and the FFT. The graph also provides a visual representation of the arm-interarm contrast, an interesting topic of study [11].

Figure 2 shows the counting function for a 2-arm galaxy with symmetrically spaced arms, 3-arm galaxy with symmetrically spaced arms, and a 3-arm galaxy with two 180° symmetric arms and one asymmetric arm. Note that SpiralArmCount sees the 3-arm galaxy with two bright 180° symmetric arms as a 2-arm spiral because of its rotational symmetry.

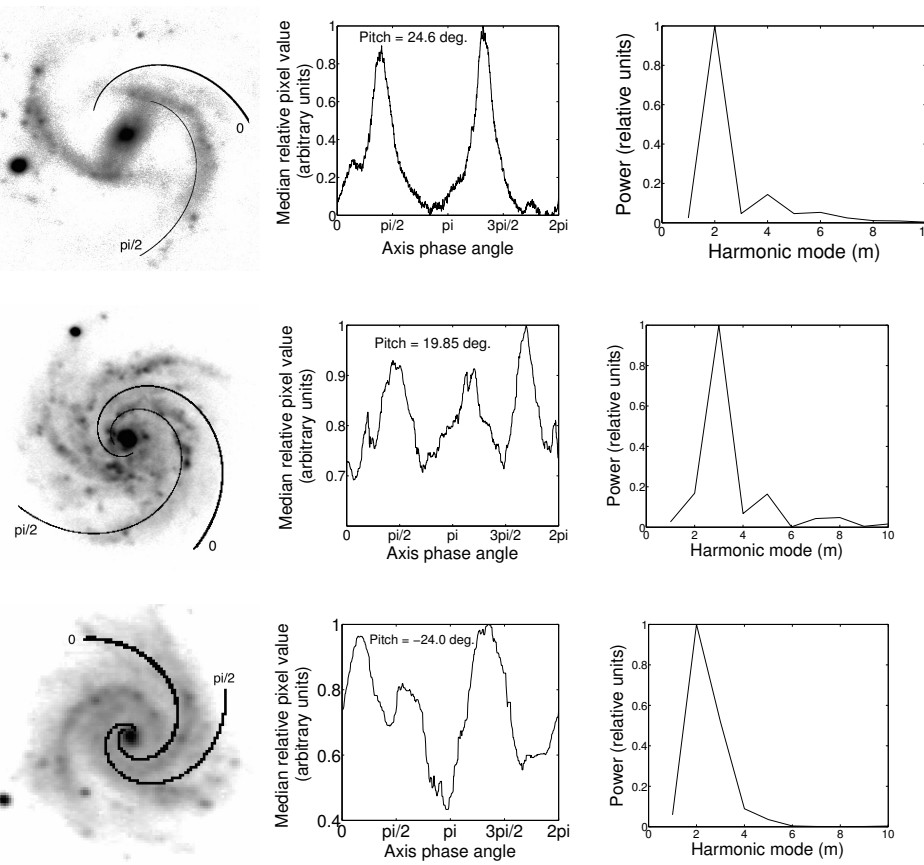

**Figure 2.** SpiralArmCount's outputs. **Top row**: UGC 1081 (*r*-band, WIYN 3.5 m) with two symmetrically spaced arms. **Middle row**: UGC 463 (*B*-band, Kitt Peak 2.1 m) with three symmetrically spaced arms. **Bottom row**: UGC 4107 (*g*-band, SDSS) with two bright arms approximately 180° apart and a third, slightly smaller arm. The **left column** shows each galaxy image, annotated with the spiral axes at respective phase angles 0 and $\pi/2$ radians. The **middle column** shows the median pixel value along each spiral axis, where the spiral coordinate system has the same pitch as the galaxy. Each local maximum represents a spiral arm. The **right column** shows the counting function. It is the FFT of the middle column, except the frequency axis is converted to harmonic modes. All images were deprojected to face-on prior to analysis.

SpiralArmCount also outputs a FITS image of the original galaxy, annotated with the spiral axes at phase angles 0 and $\pi/2$ radians. This allows the user to associate each peak in the counting function with its corresponding spiral arm. Unlike in Figure 2, SpiralArmCount does not label the two axes. Rather, the zero axis is both wider and brighter (i.e., has higher pixel values) than the $\pi/2$ axis.

## 3. Results

In this section, we present examples of pitch angle measurements on synthetically generated logarithmic spirals of known pitch, as well as on real galaxies. For synthetic spirals, the aim is to test the robustness of the algorithm by varying both the input parameters and the spiral properties. We then test the code's reliability on real galaxies by comparing its results to 2DFFT measurements of the same galaxies. Finally, we test the galaxy parameters under which the code is reliable by comparing measurements of high-resolution galaxy images to low-resolution images of the same galaxies.

### 3.1. Pitch Angle Measurement Examples

#### 3.1.1. Synthetic Spiral

Figure 3 shows Spirality's measurement of a synthetic two-arm spiral of radius 100 pixels, arm thickness 3 pixels, and pitch angle 20°. The spiral was generated by GenSpiral. The inner radius of the measurement annulus varied from 5 pixels to 65 pixels in steps of 10 pixels. The outer radius remained constant at 98 pixels. Using the variable inner radius method to establish error bars, the resulting pitch is $19.97° \pm 0.13°$.

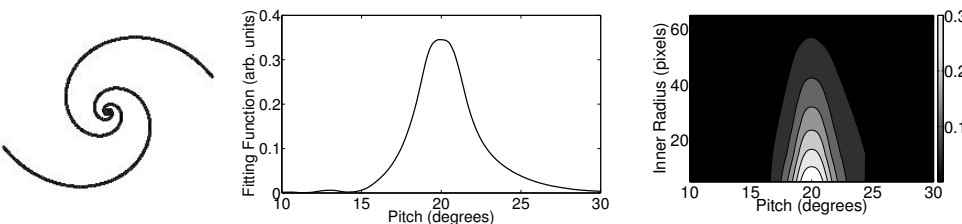

**Figure 3.** Spirality's measurement of the synthetic two-arm spiral of pitch angle 20°, shown in the **left panel**. **Center**: Fitting function vs. pitch angle for a measurement annulus with inner radius 5 pixels, or 5% of the outer radius. The outer radius of the measurement annulus is approximately equal to the outer radius of the spiral. **Right**: Fitting function vs. pitch angle and measurement annulus inner radius. Note that the center panel is a cross section of the right panel. Spirality's measurement gives a best-fit pitch of $19.97° \pm 0.13°$.

#### 3.1.2. Simple Galaxy: UGC 463

Figure 4 shows Spirality's measurement of galaxy UGC 463. This *B*-band image was taken with the 2.1-meter telescope at Kitt Peak. The inner radius of the measurement annulus varied from 0 to 45 pixels. Under normal circumstances, the outer radius would be placed at the visible edge of the spiral arms. However, this image contains a foreground star near the edge of the galaxy. In order to prevent Spirality from interpreting the star as part of a spiral arm, the outer edge of the measurement annulus was placed just inside the star's radial position. Spirality therefore did not see the star.

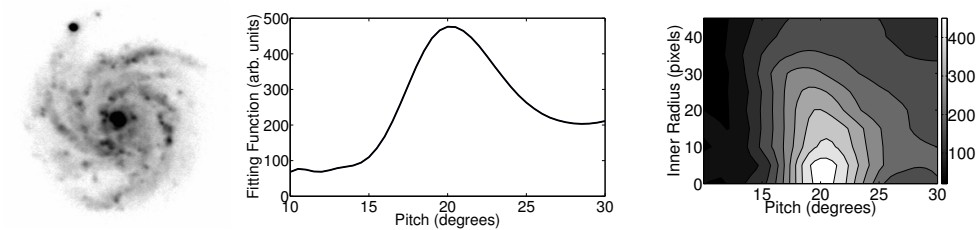

**Figure 4.** Spirality's measurement of the galaxy UGC 463, (*B*-band, Kitt Peak 2.1 m) shown in the **left panel**. **Center**: Fitting function vs. pitch angle for a measurement annulus with inner radius 5 pixels, or 4% of the galaxy's visible radius. The outer radius of the measurement annulus is 115 pixels, or 96% of the galaxy's visible radius. **Right**: Fitting function vs. pitch angle and measurement annulus inner radius. Note that the center panel is a cross section of the right panel. Spirality's measurement gives a best-fit pitch of $19.85° \pm 1.57°$, which is consistent with 2DFFT's measurement of $22.38° \pm 3.21°$.

Spirality's measurement gives a best-fit pitch of $19.85° \pm 1.57°$. For comparison, 2DFFT's measurement of the image's 3-arm component is $22.38° \pm 3.21°$. Visual inspection, done by overlaying transparencies marked with spirals of known pitch onto the spiral arms, suggests a pitch of $20° \pm 5°$

With an error bar of only $1.57°$, Spirality is particularly confident in this measurement. There are several reasons that combine to make this galaxy easy to measure. First, the absence of foreground stars in the inner part of the image allows Spirality to fit the entire galaxy, not just the 2-arm symmetric component. Because the spiral arms extend from the

central region to the outer edge of the galaxy, Spirality is able to track long segments of the arms. Because the spiral arms are bright, Spirality has no difficulty distinguishing the arms from the space in between. Finally, like many (though certainly not all) spiral galaxies, UGC 463 has arms that are nearly logarithmic, i.e., the pitch does not change much along the radial coordinate.

### 3.1.3. Interesting Galaxy: UGC 4256

Occasionally there exists a galaxy with different spirals of different pitch angles. Figure 5 shows one such galaxy. UGC 4256 consists of a single, bright, tail-like spiral with a pitch angle of about 45°. The bright spiral overlays two dimmer symmetric spirals with pitch angles of about 30°. This *B*-band image was taken with Jacobus Kapteyn Telescope (JKT).

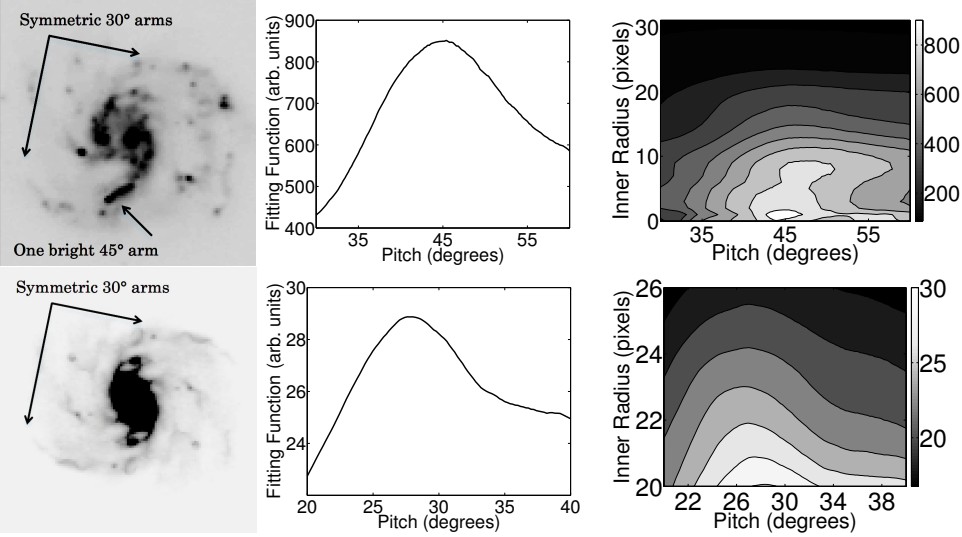

**Figure 5.** Spirality measurements of the galaxy UGC 4256, both the entire galaxy (**top row**) and the 2-arm component (**bottom row**). Top row, left to right: *B*-band image of the entire galaxy, fitting function vs. pitch angle at a fixed inner radius, fitting function vs. both pitch angle and inner radius. Bottom row: Same as top row, but for the galaxy's 2-arm (180° rotationally symmetric) component, which was computed by SymPart. Note that in each row, the center panel is a cross section of the right panel. For the whole galaxy, Spirality yields a measurement of $45.1° \pm 3.8°$, which is consistent with transparency overlay for the single bright spiral. For the 2-arm component, Spirality yields a measurement of $27.2° \pm 4.1°$, which is consistent with 2DFFT's $29.1° \pm 4.3°$ measurement of the 2-arm mode.

When Spirality measures the galaxy as a whole, it fits the single bright spiral to a template of $45.1° \pm 3.8°$. This measurement is consistent with the result from fitting the arm to spirals of known pitch using transparency overlays. 2DFFT, which sometimes produces unreliable results in the 1-arm mode, is not able to see this spiral.

On the other hand, when Spirality measures only the 2-arm symmetric component, the single bright spiral disappears, and the dimmer symmetric spirals come to the fore. Spirality fits the symmetric arms to a template of $27.2° \pm 4.1°$, which is consistent with 2DFFT's 2-mode measurement of $29.1° \pm 4.3°$, and also with the result from transparency overlays.

### 3.2. Tests on Synthetic Spirals

We conducted extensive tests on synthetically generated spirals in order to determine which regimes Spirality excels and in which regime a different method might be preferable. These spirals are variants of what we call the "Ideal Spiral", a noiseless, logarithmic, synthetically generated spiral with two symmetric arms, pitch angle 20°, radius 100 pixels, arm thickness 4 pixels, and a face-on orientation, with no bar or bulge. We tested 77 synthetic

spirals with properties that varied from the Ideal Spiral. Additionally, we performed 40 tests on the Ideal Spiral with varying input parameters.

### 3.2.1. Varying Spiral Properties

These synthetic spirals varied from the Ideal Spiral in the number of spiral arms, pitch angle, degree of logarithmicity, radius, signal-to-noise ratio (SNR), inclination angle, bar length, and bulge radius, respectively. The results are summarized in Table 1. The table's rows are discussed below.

**Table 1.** Test results for variations of the ideal [1] synthetic spiral. Each triplet of rows shows a variation on one of the spiral's properties. For each triplet, the top row shows the value of the quantity being varied, the middle row shows the measured pitch in degrees, and the bottom row shows the measurement uncertainty. For Spirality's input parameters, we used the so-called optimal inputs described in Table 2.

| No. of spiral arms | 1 | 2 | 3 | 4 | 5 | 6 | 8 |
|---|---|---|---|---|---|---|---|
| $P$ | 20.01 | 20.01 | 20.00 | 20.01 | 20.00 | 20.00 | 20.01 |
| $\pm$ | 0.12 | 0.12 | 0.10 | 0.11 | 0.10 | 0.10 | 0.11 |
| True pitch (°) | 5 | 10 | 15 | 25 | 30 | 40 | 50 |
| $P$ | 5.02 | 10.03 | 14.96 | 24.87 | 30.23 | 40.05 | 49.62 |
| $\pm$ | 0.12 | 0.14 | 0.12 | 0.48 | 0.29 | 0.76 | 0.72 |
| Non-logarithmicity [2] (°) | −19 | −15 | −10 | −5 | 5 | 10 | 15 |
| $P$ | 8.41 | 10.20 | 13.26 | 16.25 | 23.43 | 26.75 | 29.99 |
| $\pm$ | 5.06 | 3.54 | 2.04 | 0.82 | 0.79 | 1.86 | 3.21 |
| Spiral radius (px) | 15 | 20 | 25 | 35 | 50 | 75 | 200 |
| $P$ | 20.73 | 20.04 | 19.66 | 19.75 | 20.02 | 20.03 | 20.00 |
| $\pm$ | 1.69 | 0.82 | 0.64 | 0.29 | 0.12 | 0.12 | 0.10 |
| Arm thickness (px) | 1 | 3 | 6 | 10 | 15 | 20 | 25 |
| $P$ | 19.99 | 19.97 | 20.00 | 20.03 | 19.72 | 20.60 | 19.48 |
| $\pm$ | 0.12 | 0.13 | 0.10 | 0.45 | 0.66 | 1.38 | 0.79 |
| $SNR$ [3] | 16 | 8 | 4 | 2 | 1 | 0.5 | 0.25 |
| $P$ | 20.10 | 20.02 | 19.96 | 20.10 | 20.05 | 20.23 | 20.86 |
| $\pm$ | 0.10 | 0.18 | 0.20 | 0.20 | 0.64 | 0.50 | 0.73 |
| Inclination (°) | 5 | 10 | 15 | 20 | 25 | 30 | 35 |
| $P$ | 20.17 | 20.30 | 20.46 | 20.74 | 20.68 | 20.63 | 21.59 |
| $\pm$ | 0.34 | 0.56 | 1.26 | 2.27 | 3.54 | 5.15 | 7.61 |
| Bar half-length (px) | 10 | 20 | 30 | 40 | 50 | 60 | 70 |
| $P_{center}$ [4] | 20.11 | 20.15 | 20.18 | 20.21 | 20.27 | 33.19 | 40.18 |
| $\pm$ | 0.18 | 0.18 | 0.18 | 0.18 | 0.21 | 14.32 | 15.70 |
| $P_{bar}$ [5] | 20.02 | 20.02 | 20.03 | 20.00 | 20.11 | 20.02 | 20.13 |
| $\pm$ | 0.14 | 0.12 | 0.13 | 0.10 | 0.15 | 0.14 | 0.19 |
| Bulge radius (px) | 10 | 20 | 30 | 40 | 50 | 60 | 70 |
| $P_{center}$ [4] | 20.06 | 20.05 | 20.04 | 20.04 | 20.09 | 20.10 | 20.10 |
| $\pm$ | 0.11 | 0.11 | 0.11 | 0.11 | 0.10 | 0.10 | 0.10 |
| $P_{bulge}$ [5] | 20.00 | 20.02 | 20.01 | 20.00 | 20.11 | 20.02 | 20.13 |
| $\pm$ | 0.10 | 0.11 | 0.11 | 0.10 | 0.15 | 0.13 | 0.20 |

[1] The ideal synthetic spiral, as defined here, has two arms, has pitch angle 20°, is logarithmic, has radius 100 pixels and arm thickness 4 pixels, contains no noise, is oriented face-on, and has no bar or bulge. [2] Linear change in pitch angle from the center to the edge. The pitch remains 20° at the center. [3] Signal-to-noise ratio, where Gaussian noise is added to the image. [4] Measurement annulus includes the entire spiral, including the bulge or bar. [5] Measurement annulus extends from the edge of the bar or bulge to the spiral rim.

**Table 2.** Pitch angle measurement results after varying the optimal [1] inputs on the so-called ideal synthetic spiral described in Table 1. Each triplet of rows shows a variation on one of Spirality's input parameters. For each triplet, the top row shows the value of the quantity being varied, the middle row shows the measured pitch in degrees, and the bottom row shows the measurement error.

| Center offset [2] (px) | 0 | 2 | 4 | 6 | 8 | 10 |
|---|---|---|---|---|---|---|
| $P$ | 20.01 | 20.01 | 19.95 | 20.12 | 19.96 | 22.03 |
| $\pm$ | 0.12 | 0.31 | 4.04 | 6.95 | 10.15 | 13.41 |
| Inner radius spacing [3] (px) | 2 | 5 | 10 | 15 | 20 | 25 |
| $P$ | 20.02 | 20.02 | 20.01 | 20.02 | 20.03 | 20.00 |
| $\pm$ | 0.11 | 0.12 | 0.12 | 0.12 | 0.13 | 0.10 |
| Number of spiral axes [4] | 10 | 50 | 100 | 500 | 1000 | 4000 |
| $P$ | 20.70 | 19.00 | 19.20 | 19.81 | 19.82 | 19.83 |
| $\pm$ | 0.10 | 0.10 | 0.21 | 0.24 | 0.16 | 0.16 |
| Pitch angle spacing [5] (°) | 0.1 | 0.3 | 0.5 | 1 | 3 | 5 |
| $P$ | 20.00 | 19.86 | 20.00 | 20.00 | 19.00 | 20.00 |
| $\pm$ | 0.10 | 0.44 | 0.50 | 1.00 | 3.00 | 5.00 |
| Axis point spacing [6] (px) | 0.25 | 0.5 | 0.75 | 1 | 1.5 | 2 |
| $P$ | 19.91 | 19.91 | 19.93 | 19.87 | 19.84 | 19.96 |
| $\pm$ | 0.12 | 0.25 | 0.15 | 0.18 | 0.20 | 0.18 |

[1] The optimal inputs are: center offset 0, inner radius spacing 10 pixels, 650 spiral axes, pitch angle spacing 0.1°, axis point spacing 0.1 pixels. [2] The number of pixels between the spiral's true center and the center of the measurement annuli. [3] The number of pixels between inner radii of successive measurement annuli. [4] Number of spiral coordinate axes computed for each pitch angle template. [5] Number of degrees between successive pitch angle templates. [6] Number of pixels between successive computation points on a given spiral axis.

### Number of Spiral Arms

The number of arms was varied from 1 to 8. Neither the accuracy nor the error bar depended on the number of arms. For all such spirals, the measured pitch was within 0.01° of the true pitch, the total error was less than 0.13°, and the error bar was consistent with the accuracy.

### Pitch

The spiral's true pitch was varied from 5° to 50°, which is the range of pitch angles for most galaxies. The accuracy was slightly reduced and the error bar grew slightly as true pitch increased. For all such spirals, the measured pitch was within 0.4° of the true pitch, the total error was less than 0.8°, and the error bar was consistent with the accuracy.

### Non-Logarithmicity

Spirals with varying deviations from logarithmicity were measured. For each spiral, the true pitch was 20° at the center. The deviation from logarithmicity $\Delta P_{true}$ is the difference between the true pitch at the outer rim of the spiral and the true pitch at the center. For example, $\Delta P_{true} = 10°$ represents a spiral that varies linearly from 20° at the center to 30° at the outer rim. Because Spirality's templates are logarithmic, we would expect the code to output large error bars for nonlogarithmic spirals. Indeed, greater values of $\Delta P_{true}$ (that is, less logarithmic spirals) yielded measurements with larger error bars. However, the measured pitch was usually consistent with the spiral's mean pitch as a function of radius. For 5 of the 7 spirals, the measured pitch was less than 1 error bar away from the mean true pitch. For each of the remaining 2 spirals, the measured pitch was less than 1.6 times the error bar away from the mean true pitch.

Radius

The spiral radius was varied from 15 pixels to 200 pixels. The error bar shrank, and the accuracy improved, as the spiral grew in radius. For all such spirals, the measured pitch was within $0.7°$ of the true pitch, the total error was less than $1.7°$, and the measured pitch was within 1.1 times the error bar away from the true pitch.

Arm Thickness

The spiral arm thickness was varied from 1 pixel to 25 pixels. The error bar grew, and the accuracy suffered, as the thickness increased. For all such spirals, the measured pitch was within $0.6°$ of the true pitch, the total error was less than $1.4°$, and the error bar was consistent with the accuracy.

SNR

Varying amounts of Gaussian noise were added to the spiral image. Here, SNR is defined as the mean pixel value of the spiral arm divided by the standard deviation of the Gaussian noise distribution. The mean pixel value of the empty space between the arms was zero. Noise was added to both the spiral arms and the empty space. The SNR varied from 16 down to 0.25. The error bar increased, and the accuracy suffered, as the SNR decreased. For all such spirals, the measured pitch was within $0.9°$ of the true pitch, the total error was less than $0.8°$, and the measured pitch was less than 1.2 times the error bar away from the true pitch.

Inclination

For a galaxy, the inclination angle would be estimated or measured, and the galaxy deprojected to face-on, before the pitch is measured. However, inclination angles can be challenging to find. Therefore it is prudent to know how much leeway a pitch angle measurement tool allows for mismeasuring the inclination angle. For this test, the synthetic spiral was compressed along the y-axis in order to simulate viewing the spiral at inclination angles varying from $5°$ (i.e., nearly face-on) to $35°$. The result is that the error bar grew substantially, but the accuracy only suffered slightly, as the inclination increased. For all such spirals, the measured pitch was within $1.6°$ of the true pitch, and the error bar was consistent with the accuracy. We find that Spirality states the correct pitch angle even if a galaxy's inclination is incorrectly deprojected. However, the measurement yields unreasonable error bars if the galaxy retains an inclination of more than $20°$ after deprojection.

Bar Half-Length

The central section of the spiral image was replaced by an elliptical bar with an axis ratio of approximately 1.7. The bar's half-length, or semi-major axis, varied up to 70 pixels. When the measurement annulus included the elliptical bar, the error bar increased substantially and the accuracy suffered substantially if the bar's half-length was more than 50 pixels (i.e., if the elliptical bar consumed more than half the radius of the spiral). For spirals with bar half-lengths of 50 pixels or less, the total error was less than $0.3°$ and the measured pitch was less than 1.3 times the error bar away from the true pitch. On the other hand, when the measurement annulus was placed outside the elliptical bar (as would be the case in a galaxy measurement), neither the error bar nor the accuracy were substantially affected by the elliptical bar. For all such measurements, the total error was less than $0.2°$ and the accuracy was consistent with the error bar.

Bulge Radius

The central section of the spiral image was replaced by a circular bulge, the radius of which varied up to 70 pixels. When the measurement annulus included the bulge, the total error was less than 0.2° and the accuracy was consistent with the error bar. When the measurement annulus was placed outside the bulge, the total error was less than 0.3° and the accuracy was consistent with the error bar.

### 3.2.2. Varying Inputs

The Ideal Spiral defined in the opening paragraph of Section 3.2 was measured with varying inputs to the Spirality code. The results are summarized in Table 2.

Center Offset (Input Variables **X0** and **Y0**)

A galaxy's center can be estimated, though the estimate contains some error. It is therefore helpful to know how much leeway a pitch angle measurement tool allows for the mismeasurement of a galaxy's center. For this test, the center offset is defined as the difference, in pixels, between the center of the spiral and the center of the measurement annulus. Each pixel represents 1% of the spiral's radius. The center offset was varied from 0 to 10 pixels. For all such measurements, the error bar was consistent with the accuracy. The error bar was highly sensitive to the center offset, though the accuracy was only mildly affected. For example, when the center offset was less than or equal to 8 pixels, the measured pitch was within 0.12° of the true pitch. However, with a center offset of 8 pixels, the error bar had already grown to more than 10°. For galaxies in which the center is difficult to obtain, we recommend varying the center coordinates of the measurement annulus until the error bar is minimized.

Inner Radius Spacing (Input Variable **Inner Radius Spacing**)

Spirality's error bars are determined using the method of Davis et al. [10], in which the pitch angle is measured on several measurement annuli. Each annulus has the same outer radius but a different inner radius. The spacing, in pixels, between successive inner radii is input. The smaller the inner radius spacing, the more annuli are measured. For this test, the inner radius spacing was varied from 2 to 25 pixels. For all such measurements, the error bar was less than 0.14°, and was consistent with the accuracy.

We recommend a two-step measurement process for galaxies. In the first measurement, which we call the "coarse" measurement, the inner radius begins just outside the bar or bulge, and is varied outward with a sufficiently small inner radius spacing to allow about 10 different inner radii between the bulge/bar and the outer radius. Once a region of inner radii of roughly constant pitch is identified, a second measurement (the "fine" measurement) is conducted. The fine measurement is "zoomed in", both on the stable radius segment and on the resulting pitch angle. In other words, the second measurement includes 10 inner radii that span only the stable region rather than the galaxy as a whole, and only look at pitch angle templates that are near the true pitch.

Number of Spiral Axes (Input Variable **NAXIS**)

The number of spiral axes on the templates was varied from 1 to 4000. When the number of spiral axes was at least 500, or 5 times the spiral's radius in pixels, the error bar was less than 0.25°, and the measured pitch was within 1.2 times the error bar away from the true pitch. For galaxy measurements, we recommend at least $4\pi R$ spiral axes, where $R$ is the galaxy's radius in pixels. That way, each pixel on the outer rim of the measurement annuli gets counted, on average, twice. If the image is sufficiently large that computation time becomes an issue, the number of spiral axes can be reduced to $2\pi R$, thus counting each outer pixel, on average, once.

Pitch Angle Spacing (Input Variable **PSTEP**)

The spacing, in degrees, between the pitch angles of successive templates is input. The pitch spacing is the so-called "quantized error" described by Davis et al. [10]. As such, it is the minimum possible error bar. For this test, the pitch spacing was varied from $0.1°$ to $5°$. For all such measurements, the measured pitch was within $0.25°$ of the true pitch, and the error bar was consistent with the accuracy. However, since the pitch spacing is also the quantized error, the error bar grew with the pitch spacing.

For the two-step galaxy measurement described above, we recommend a pitch spacing of $1°$ for the course measurement and $0.2°$ for the fine measurement. The fine measurement should be zoomed into a pitch angle domain on which the fitting function's peak is a local max. If the quantized error contributes significantly to the overall error, the code will generate a warning. The solution is to reduce the pitch spacing.

Axis Point Spacing (Input Variable **Axis Point Spacing**)

The spacing, in pixels, between successive computation points on a given spiral axis is input. For this test, the axis point spacing was varied from 0.25 pixels to 2 pixels. When the axis point spacing was 1.5 pixels or less, the measured pitch was within $0.16°$ of the true pitch and the error bar was consistent with the accuracy. For galaxy measurements, we recommend an axis point spacing of at most 0.25 pixels. If the image is sufficiently large that computation time becomes an issue, the axis point spacing may be increased to 0.5 pixels.

*3.3. Tests on Galaxy Samples*

We tested Spirality on three samples of galaxies: the nearby sample defined by the DMS PPak [26], the nearby sample defined by Miller et al. [22], and a distant sample of visually identified spirals in GOODS North and South [27].

Tests were conducted by comparing Spirality pitch angle measurement results with those produced by 2DFFT [28], and also by overlaying the galaxies with spirals of known pitch via transparencies. In some cases, where high-resolution images and low-resolution images were available for the same galaxy. In each such case, Spirality was tested by comparing its measurement of the high resolution image to its measurement of the low-resolution image.

For these tests, we define the "disagreement factor" as the difference between the two codes' results divided by the sum of the error bars. If the disagreement factor is less than one, Spirality's measurement is consistent with 2DFFT's measurement. If the disagreement factor is equal to one, Spirality's error bar barely touches 2DFFT's error bar. If the disagreement factor is greater than one, the error bars do not overlap.

3.3.1. Nearby Galaxies: DMS PPak

The sample defined by DMS PPak [26] contains 30 nearby galaxies. Spirality's pitch angle measurements are consistent with 2DFFT's pitch angle measurements for 26 of those galaxies, as shown in Table 3.

The four galaxies for which Spirality disagrees with 2DFFT are discussed in the subsections below.

**Table 3.** Spirality vs. 2DFFT pitch angle measurements for nearby galaxies.

| Galaxy Name | Type [1] | Band | Source [2] | Spirality Pitch (°) | 2DFFT Pitch [3] (°) | D [4] |
|---|---|---|---|---|---|---|
| UGC 448 | SABc | *r* | a | $-15.1 \pm 4.9$ | $-18.1 \pm 1.7$ | 0.46 |
| UGC 463 | SABc | *B* | d | $19.9 \pm 1.6$ | $22.4 \pm 3.2$ | 0.52 |
| UGC 1081 | SBc | *r* | a | $24.6 \pm 2.0$ | $24.3 \pm 3.1$ | 0.06 |
| UGC 1087 | Sc | *r* | a | $9.7 \pm 5.1$ | $10.6 \pm 2.2$ | 0.13 |
| UGC 1529 | Sc | *B* | d | $-28.3 \pm 3.3$ | $-26.1 \pm 4.4$ | 0.28 |
| UGC 1635 | Sbc | *r* | a | $9.3 \pm 1.5$ | $11.8 \pm 0.8$ | 1.09 |
| UGC 1862 | SABcd [5] | *r* | a | $27.4 \pm 8.1$ | $23.9 \pm 3.5$ | 0.30 |
| UGC 1908 | SBc [6] | *B* | d | $22.4 \pm 1.1$ | $20.6 \pm 3.5$ | 0.39 |
| UGC 3091 | SABd | *i* | a | $-14.6 \pm 5.9$ | $-29.5 \pm 4.0$ | 1.50 |
| UGC 3140 | Sc | *r* | a | $-19.7 \pm 1.8$ | $-16.2 \pm 4.8$ | 0.54 |
| UGC 3701 | Scd | *r* | a | $-14.8 \pm 4.7$ | $-15.4 \pm 4.8$ | 0.07 |
| UGC 3997 | Im | *g* | b | $-16.2 \pm 2.5$ | $-10.5 \pm 2.6$ | 1.12 |
| UGC 4036 | SABbc | *B* | d | $-16.9 \pm 4.1$ | $-15.0 \pm 1.1$ | 0.36 |
| UGC 4107 | Sc | *g* | b | $-24.3 \pm 3.1$ | $-20.4 \pm 2.1$ | 0.76 |
| UGC 4256 | SABc | *g* | b | $45.1 \pm 3.8$ | $29.1 \pm 4.3$ | 1.97 |
| UGC 4368 | Scd | *g* | b | $34.3 \pm 6.2$ | $23.7 \pm 2.1$ | 1.28 |
| UGC 4380 | Scd | *g* | b | $-15.4 \pm 4.3$ | $-23.3 \pm 4.6$ | 0.89 |
| UGC 4458 | Sa | *g* | b | $-9.7 \pm 4.0$ | $-13.6 \pm 3.0$ | 0.56 |
| UGC 4555 | SABbc | *g* | b | $12.6 \pm 0.5$ | $12.1 \pm 1.0$ | 0.38 |
| UGC 4622 | Scd | *g* | b | $-15.1 \pm 3.3$ | $-21.8 \pm 4.9$ | 0.83 |
| UGC 6903 | SBcd | *g* | b | $-14.8 \pm 2.1$ | $-15.8 \pm 2.2$ | 0.23 |
| UGC 6918 | SABb [7] | F606W | c | $-15.2 \pm 1.2$ | $-17.0 \pm 2.3$ | 0.53 |
| UGC 7244 | SBcd | *g* | b | $25.7 \pm 11.6$ | $32.1 \pm 4.3$ | 0.40 |
| UGC 7917 | SBbc | *g* | b | $-14.2 \pm 4.9$ | $-15.5 \pm 1.4$ | 0.21 |
| UGC 8196 | Sb | *g* | b | $-7.3 \pm 1.6$ | $-8.2 \pm 0.5$ | 0.42 |
| UGC 9177 | Scd | *g* | b | $-12.7 \pm 2.9$ | $-14.4 \pm 1.9$ | 0.35 |
| UGC 9837 | SABc | *g* | b | $28.6 \pm 4.6$ | $25.7 \pm 2.8$ | 0.39 |
| UGC 9965 | Sc | *g* | b | $-12.7 \pm 2.2$ | $-13.3 \pm 2.0$ | 0.15 |
| UGC 11,318 | SBbc | *B* | d | $-34.7 \pm 2.9$ | $-29.7 \pm 4.4$ | 0.70 |
| UGC 12,391 | SABc | *r* | a | $-11.3 \pm 0.8$ | $-13.2 \pm 5.0$ | 0.33 |

[1] Hubble morphological type from either the UGC [29] or RC3 [30] catalog. Morphology notes: 5 = peculiar, 6 = starburst, 7 = AGN. [2] Image sources: (a) WIYN 3.5 m pODI, (b) SDSS, (c) HST, and (d) Kitt Peak 2.1 m. [3] 2DFFT pitch angle measurements from Davis et al. [31]. [4] Disagreement factor, or the difference between measurements divided by the sum of the uncertainties.

UGC 1635

The disagreement factor for this galaxy was 1.09. Although the error bars did not quite overlap, the Spirality's measurement was very nearly consistent with 2DFFT's. Because our error bars are computed from standard deviations, they are based on the assumption that the best-fit pitch follows Gaussian statistics. We would therefore expect slight disagreements among measurements a small fraction of the time.

UGC 3091

With a disagreement factor of 1.5, Spirality's result disagrees significantly with 2DFFT's result. The respective error bars of 5.9° and 4.9° show that neither code is particularly confident in its measurement. This galaxy has a low surface brightness, and its spiral structure is difficult to make out by visual inspection. Such galaxies pose a challenge to any pitch angle measurement code.

UGC 3997

This is a particularly interesting case. When one of our researchers measured this galaxy using Spirality, and another using 2DFFT, without communicating with one another, the researchers reached a disagreement factor of 1.12, as indicated in Table 3. Afterward,

when the researchers compared results, they found that 2DFFT had been instructed to measure the spirals in the outer disk, while Spirality had been instructed to measure the spirals in the inner disk. When Spirality's measurement annulus was adjusted to match 2DFFT's, Spirality produced a result of $-12.4° \pm 0.9°$, which is consistent with 2DFFT's result.

This case serves as a cautionary tale for any pitch angle measurement algorithm that assumes a logarithmic spiral, i.e., that declares a best-fit pitch. Such a declaration can be useful because global spiral arms are often well-described as logarithmic. However, when assuming a constant pitch and thus reducing the spiral structure to a single quantity, a measurement annulus must be declared. Whether the annulus is determined by inspection or by analysis, the resulting best-fit pitch may well depend on the annulus. For galaxies in which the structure is too complex to be described by a single quantity, researchers may choose to analyze individual spiral arms or spiral arm segments. Such analysis can be done using methods such as SpArcFiRe, which was introduced in Davis and Hayes [13].

UGC 4256

This galaxy is discussed in Section 3.1.3 and shown in Figure 5. The disagreement factor of 1.97 stems from 2DFFT's assumption of 2-arm symmetry. 2DFFT only measures the dimmer but 180° symmetric pair of arms with pitch angle 29.1°. Spirality, on the other hand, measured the brighter single arm with pitch angle 45.1° The disagreement is resolved when Spirality is directed to measure only the 2-arm component.

### 3.3.2. Measurement Quality of Low-Resolution Images

We have found that measurement quality depends strongly on the radius, in pixels, of the galaxy image.

For many of the galaxies listed in Table 3, we had access to both high-resolution imaging and low-resolution imaging. For each galaxy where such imaging was available, we used Spirality to measure the pitch angle of the low-resolution image, and also of and the high-resolution image, and compared the results. We declared each result a "confirmed" measurement if the result was consistent with Spirality's measurement of the same galaxy at a different resolution, or else if the result was consistent with 2DFFT's measurement of the same image.

For the 30 galaxies in Table 3, a total of 44 images were analyzed. 23 of those images had galactic radii of 30 pixels or less, while the remaining images had galactic radii of 35 pixels or more.

For galaxies of radius $\geq 35$ pixels, 100% of the pitch angle measurements were confirmed either by higher resolution imaging or by an independent method of measurement. For galaxies of radius $\leq 30$ pixels, only 43% of the measurements were confirmed.

This result underscores the importance of having adequate image resolution if a pitch angle measurement is to be trusted.

### 3.3.3. More Nearby Galaxies: Pitch Angle vs. Wavelength

Miller et al. [22] measured 40 nearby spirals in three wavelength bands: *B* (new stars), 8.0 μm (starforming dust) and 3.6 μm (old stars). All measurements were taken using both Spirality and 2DFFT, shown in Figure 6.

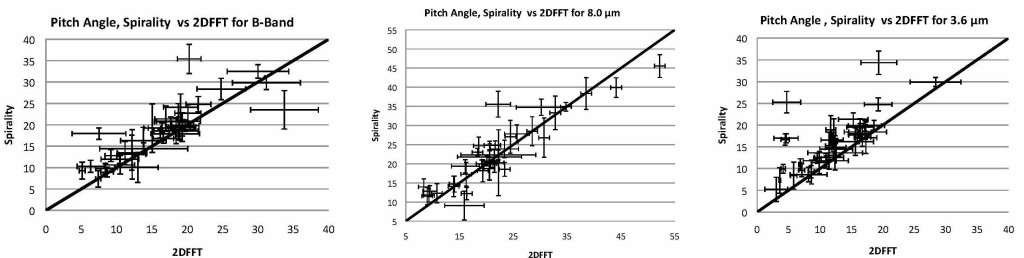

**Figure 6.** Spirality vs. 2DFFT measurements of 40 nearby spiral galaxies from Spitzer. Measurements were taken in three wavelength bands: 445 nm (*B*-Band, **left**), 3.6 μm (**center**) and 8.0 μm (**right**). In each panel, the diagonal line is $y = x$.

In the *B*-band and 3.6 μm band, the pitch angle results are mostly consistent. However, in those galaxies where the results are not consistent, Spirality shows a systematic tendency toward looser pitch angles. The average difference between the two codes' results is 4.0° for 3.6 μm and 3.1° for *B*-band.

In the 8.0 μm images the average difference is 3.6° and there is no systematic bias.

We infer that the 8.0 μm band, which tracks dust, provides smoother spiral arms and is thus easier for both codes to measure. The other two bands, which track young stars (*B*-band) and old stars (3.6 μm), provide clumpier spirals, which tend to be read as looser by Spirality than by 2DFFT.

The total average difference is 3.5° for all 40 galaxies, but is reduced to 1.8° when we only consider the 17 unbarred spirals.

### 3.3.4. High Redshift, Low Resolution Galaxies

Measuring the pitch angles of high-redshift galaxies represents a unique challenge. Many spirals in the GOODS sample, for example, have small angular radii and are quite noisy. Moreover, there is debate as to whether such galaxies even exhibit *bona fide* spiral structure. Guo et al. [32] argue that many galaxies at $z \geq 0.5$ exhibit large clumps of stars rather than logarithmic spirals. It would be no surprise if Spirality and 2DFFT, which both assume logarithmic spirals, would have difficulty coming to agreement on the pitch angle of a galaxy which may in fact be nowhere near logarithmic.

Indeed, the low resolution of distant GOODS galaxies was a key motivation for developing Spirality. We wanted to find an independent pitch angle measurement technique that would serve as a comparison to 2DFFT's results. If the two methods agree on a galaxy's pitch angle, then we gain confidence that the galaxy approximates a logarithmic spiral.

Figure 7, however, shows a particularly difficult example. In addition to being distant ($z = 1.2$) and having a small pixel radius (30 px), the galaxy also has a visual companion that is not shown in the figure. The visual companion evokes the suspicion that the disk may be distorted and the galaxy may not be logarithmic at all. It is not surprising, then, that Spirality and 2DFFT differ wildly on the results of this pitch angle. Their respective measurements are $-30.2° \pm 4.0°$ and $+37.9° \pm 6.7°$, which amounts to a disagreement of $D = 6.3$. Such a galaxy would be deemed to have an unreliable pitch angle for the purposes of scientific analysis.

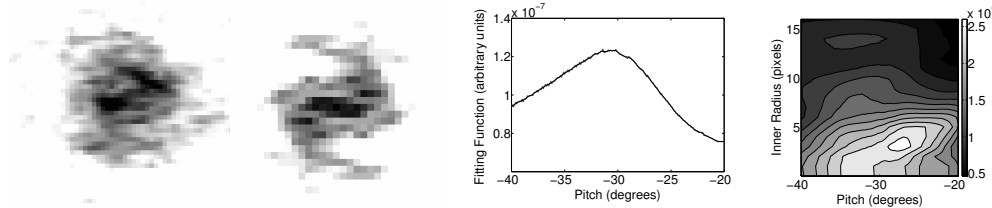

**Figure 7.** Spirality's measurement of a low-resolution galaxy from the GOODS sample. The **left panel** shows the entire galaxy, while the **second panel** show the 2-arm symmetric mode, as computed by Spirality. The **third panel** shows the pitch angle measurement of the 2-arm mode at a fixed inner radius, and the **right panel** shows the measurement with variable inner radius. No reasonable measurement could be obtained for the galaxy as a whole; the right two panels describe only the 2-arm mode. Spirality's measurement of the 2-arm mode is $-30.2° \pm 4.0°$, as compared to 2DFFT's measurement of $+37.9° \pm 6.7°$. Such a wide disagreement would cause us to expel this galaxy from any sample that is used for scientific analysis.

Figure 8 shows the pitch angle measurements of 203 visually selected spirals from the GOODS sample using both Spirality and 2DFFT. It should be noted that galaxies with high disagreement factors are included here. The two codes agree on chirality (the sign of the measurement) for 94% of galaxies. The measurements agree within their error bars on 64% of the galaxies. When the codes disagree, Spirality on average sees a tighter (closer to zero) pitch angle than 2DFFT.

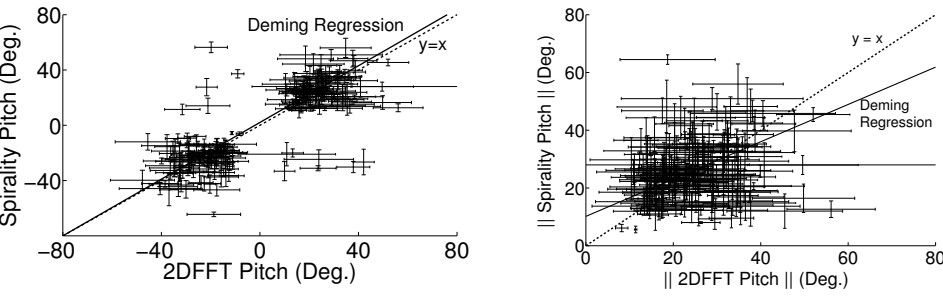

**Figure 8.** Spirality measurements vs. 2DFFT measurements of visually selected spirals in the GOODS North and South samples. For each plot, the solid line represents the orthogonal least squares (Deming) regression, while the dashed line represents $y = x$. **Left panel**: all 203 visually selected spirals. **Right panel**, absolute values for the 191 galaxies for which Spirality and 2DFFT agree on chirality.

In order to take the error bars into account when computing the regression, we expressed each data point as a bivariate Gaussian cloud of 100 randomly generated points. Each cloud is centered at the point $(P_{2D}, P_{Sp})$, where $P_{2D}$ is the 2DFFT measurement and $P_{Sp}$ is the Spirality measurement. The horizontal distribution of points for each cloud is normal, with the standard deviation being the error in the 2DFFT measurement. Likewise, the vertical distribution of points for each cloud is also normal, with the standard deviation being the error in the Spirality measurement.

The 20,300 total cloud points (100 for each of 203 galaxies) were regressed using an orthogonal (Deming) least squares fitting, as described in Kermack and Haldane [33] and revisited in York [34]. With this method, the error in the slope decreases as $\frac{1}{\sqrt{N}}$, where $N$ is the number of data points. Since our Gaussian clouds artificially increased $N$ by a factor of 100, we multiplied the resulting slope error by 10 to compensate. It should be noted that the error bars, not the Gaussian clouds, are shown in Figure 8.

The left panel of Figure 8 shows all 203 galaxies, with chirality information (positive or negative) left in tact. The slope of the Deming regression, $1.06 \pm 0.66$, is highly consistent with $y = x$, which illustrates the degree to which the two codes agree on chirality. The

intercept, $(-1 \pm 20)°$, is consistent with zero, though the large error bar illustrates the high amount of random disagreement between the two codes for this sample of high-redshift, low-resolution galaxies.

The right panel of Figure 8 shows only those 191 galaxies for which Spirality agrees with 2DFFT about the chirality. This plot shows the absolute values of those measurements. The slope of the Deming regression, $0.63 \pm 0.31$, is not consistent with $y = x$. This indicates that for poorly resolved galaxies such as these, Spirality tends to see a tighter (closer to zero) pitch than 2DFFT. The intercept is $(10 \pm 24)°$.

For this noisy sample of images, Spirality is more confident about its measurements than 2DFFT. Spirality's error bars for the GOODS sample have a mean of $4.8°$ and a standard deviation of $2.6°$. 2DFFT's error bars have a mean of $7.1°$ and a standard deviation of $4.4°$.

### 3.3.5. Testing the Error Bars

One would expect a random error distribution to be distributed more or less normally. The respective error bar distributions of Spirality and 2DFFT for the GOODS sample measurement are shown in Figure 9.

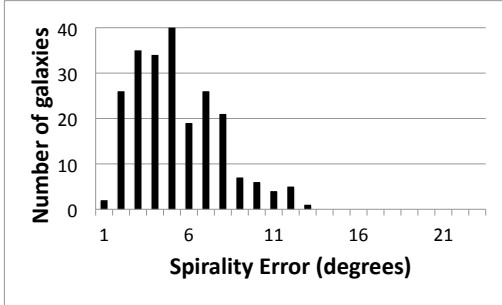 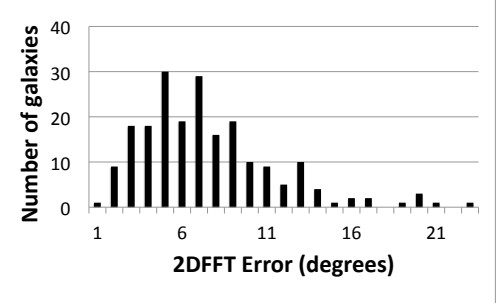

**Figure 9.** Error bar distributions for GOODS North and South pitch angle measurements of 203 galaxies by Spirality (**left**) and 2DFFT (**right**). In the right panel, a single outlier data point corresponding to a 2DFFT error of $45°$ is omitted.

Based on the Cramèr-von Mises test, the null hypothesis that Spirality's errors are distributed normally cannot be rejected at the 5% level. The test yields a *p*-value of 0.10. On the other hand, based on the same test, the null hypothesis that 2DFFT's errors are distributed normally is rejected. The test yields a *p*-value of 0.0015.

By measuring the same galaxies using both 2DFFT and Spirality, we can begin to understand the regimes in which the codes are likely to agree within their error bars. In regimes where the codes agree, we gain confidence in the measurements.

Recall from Section 3.3.1 that the disagreement factor $D$ is the difference between the Spirality measurement and the 2DFFT measurement, divided by the sum of the error bars. Therefore, if $D \leq 1$ for a galaxy, the codes agree on the galaxy's pitch angle within the error bars. While the two codes agree to within a disagreement factor of 1.1 for 90% of the nearby galaxies in the sample defined by the DMS PPak, they only agree 64% of the time in high-redshift, low-resolution images of GOODS North and South.

In order to get an idea of which galaxy properties (or combinations of properties) produce easily measurable pitch angles, we produced a series of 3-D histograms that showed the number of galaxies vs. disagreement factor and one other property (or a combination of two other properties). The histograms are shown in Figure 10.

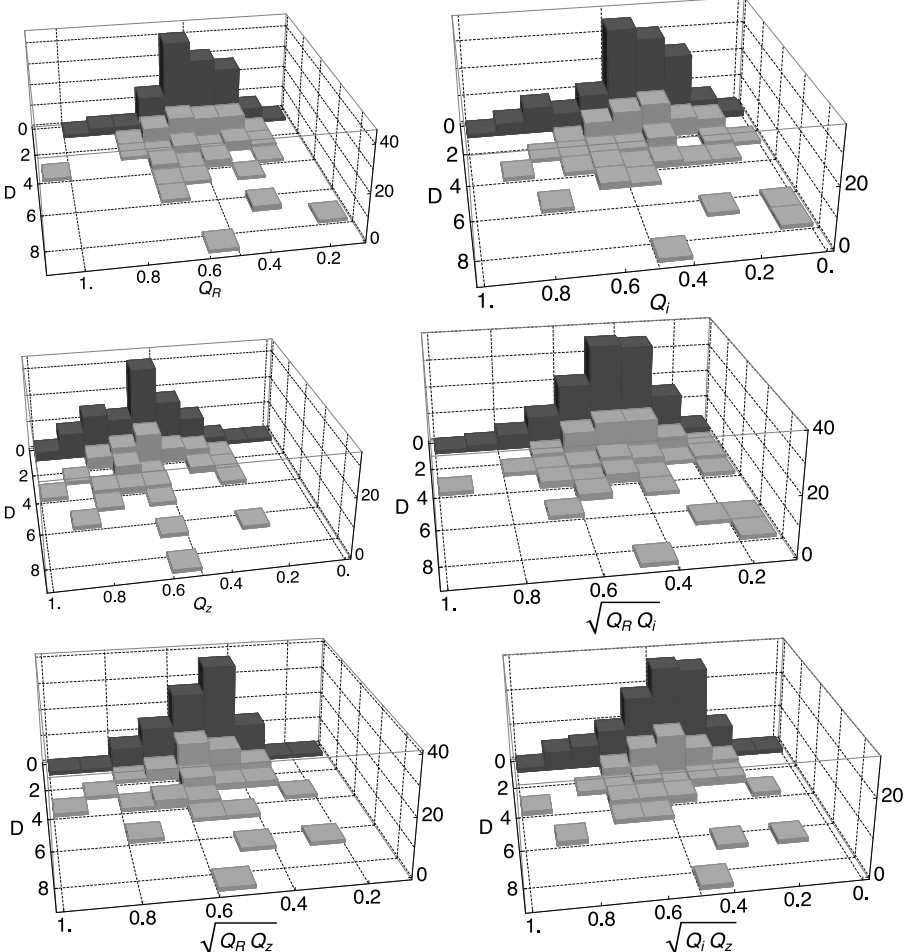

**Figure 10.** Histograms showing the disagreement factor (difference in the measurements divided by the sum the of errors) between Spirality and 2DFFT for galaxies in GOODS North and South. The disagreement factor $D$ indicates the trustworthiness of the codes' error bars. In each panel, dark gray histogram bars show galaxies for which $D < 1$, i.e., for which Spirality's pitch angle measurement is consistent with 2DFFT's measurement within the error bars. The light gray histogram bars show galaxies for which the codes disagree. The second independent variable is different for each panel. **Top left:** Radius quality, which is the galaxy's angular radius in pixels, scaled logarithmically such that its value for the largest galaxy is approximately unity. **Top right:** Brightness quality, which is the *i*-band magnitude, scaled linearly such that its value for the brightest galaxy is approximately unity and for the dimmest galaxy is approximately zero. **Middle left:** Distance quality, which is the redshift scaled linearly such that its value for the nearest galaxy approaches unity and for the most distant galaxy approaches zero. **Middle right:** Geometric mean of radius quality and brightness quality. **Bottom left:** Geometric mean of radius quality and distance quality. **Bottom right:** Geometric mean of brightness quality and distance quality.

The independent properties under analysis in Figure 10 are galaxy radius in pixels, *i*-band magnitude, and redshift. These properties are vastly different in numerical scale, so in order to combine them, we scaled each of them as a so-called quality factor from approximately zero (low quality) and approximately unity (high quality). That way, the properties can be combined as a geometric mean, and the result will also be a quality factor between zero and unity. We chose the geometric mean instead of the simple product because the simple product would combine two equal values to produce a smaller value, while the geometric mean combines two equal values to produce the same value.

The respective quality factors for radius, brightness, and redshift are:

$$Q_R = \frac{1}{3.5} ln\left(\frac{R}{270}\right) + 1 \tag{5}$$

$$Q_i = \frac{25 - i}{8} \tag{6}$$

and

$$Q_z = \frac{1.7 - z}{1.7} \tag{7}$$

where $R$ is the galaxy radius in pixels, $i$ is the $i$-band magnitude, and $z$ is the redshift.

Figure 10 shows that agreement between between Spirality and 2DFFT correlates to some, but not all, combinations of quality factors. First, the bin with the lowest brightness quality $Q_i$ shows more disagreement than agreement. This confirms the intuitive result that a dim (or, equivalently, noisy) galaxy image is difficult to measure. Second, when the combination $\sqrt{Q_R Q_i}$ of brightness quality and radius quality is low, disagreement is likely. We therefore infer that the outer regions of the disk, being generally dimmer than the inner regions, are more likely to be lost in the noise. In other words, image noise decreases the effective pixel radius on which the galaxy can be measured. Not surprisingly, disagreement shows a correlation with outer radius: the larger the galaxy, the more likely the codes are to agree. This is in line with observations of high-quality images (Section 3.3.2).

It should be noted that these quality factors are sample-dependent. No conclusion should be made about the codes' ability to agree on specific apparent magnitudes because the SNR of a galaxy image depends on the optics of the telescope and the exposure time in addition to the magnitude. In this sample, all galaxies are captured with the same optics (HST ACS), and all galaxies have exposure times of about 5 days.

The surprising result is that the disagreement factor does not correlate strongly with redshift. The difficulty the codes had in measuring galaxies in the GOODS field is therefore due more to image quality than to redshift. This is consistent with logarithmic spiral structure being fully formed in many galaxies by $z = 1.2$, the distance inside which 95% of our sample lies.

## 4. Discussion

The primary advantage of Spirality over 2DFFT is that it does not assume a specific number of spiral arms. 2DFFT, on the other hand, may yield different results for the 2-arm mode, the 3-arm mode, etc., and the user must choose the most meaningful mode. Sometimes, e.g., for a grand design 2-arm spiral, the choice of modes is obvious, and 2DFFT provides quantifiable methods to confirm the user's decision. Other times, e.g., for a flocculent galaxy, no single mode may dominate. The act of choosing a mode requires the user to discard the other modes, which may include nontrivial information. While Spirality has the option to focus on a single mode, by default it analyzes the whole of the galaxy, without throwing away any information.

Additionally, the output of Spirality's component code SpiralArmCount provides a clear way to count the spiral arms without assuming rotational symmetry, and also provides a visual representation of arm-interarm contrast.

The advantage of 2DFFT's mode decomposition, on the other hand, is that foreground stars generally have no conjugate counterparts, and therefore get stored in the 1-arm mode. Therefore, except in the unusual case of a 1-arm galaxy, 2DFFT is less sensitive to foreground stars than Spirality. In fact, the presence of even a single bright foreground star can introduce a monotonic background function in Spirality's fitting function, causing the best-fit pitch to occur at a local maximum rather than a global maximum.

In general, 2DFFT requires less computation time than Spirality. Once the galaxy has been deprojected, 2DFFT performs its analysis in a fraction of a minute, while Spirality performs its analysis in one or two minutes. The precise computation time depends on the

size of the image file, the available working memory, and the user's inputs. We recommend at least 8 GB of working memory for optimal computation time.

In general, we recommend Spirality when foreground stars cannot be subtracted. We recommend 2DFFT for very large image files, when computation time is an issue. Neither code is recommended for galaxies that are less than 30 pixels in radius.

*Future Work*

We are working to desensitize Spirality to foreground stars by changing the fitting function from the variance of means to the variance of medians. As a statistical indicator, the median is much less sensitive to outliers than the mean. In context of a galaxy image, a foreground star is an outlier in terms of pixel value. Therefore, we believe that the variance of medians will be less sensitive to foreground stars than the variance of means. Initial tests have supported this hypothesis.

Additionally, we are working to generalize Spirality so that it does not assume a logarithmic spiral. Once the best-fit pitch is determined, the corresponding logarithmic spiral template will be used as a starting point, and templates computed in which the pitch angle varies with the radial coordinate. The result will not be a best-fit global pitch, but rather a best-fit pitch as a function of radius.

**Author Contributions:** Conceptualization, D.S.; Formal analysis, D.S., B.B., C.P., B.L.D., M.H., R.M. and M.S.A.; Funding acquisition, B.L.D. and J.K.; Methodology, D.S.; Software, D.S., B.L.D. and Z.S.; Supervision, D.K. and J.K.; Visualization, D.S. and B.L.D.; Writing—original draft, D.S.; Writing—review & editing, D.S., B.B., C.P., B.L.D., M.H., R.M., Z.S., M.S.A., D.K. and J.K. All authors have read and agreed to the published version of the manuscript.

**Funding:** This research was funded in part by NSF REU Site Award 1157002. The Australian Research Council's funding scheme DP17012923 supported this research. This material is based upon work supported by Tamkeen under the NYU Abu Dhabi Research Institute grant CAP[3].

**Data Availability Statement:** The Spirality code package is written in MATLAB and can be found here: https://github.com/DeannaShields/Spirality (accessed on 1 September 2022). See the Appendix A for more detail.

**Acknowledgments:** We gratefully acknowledge Ivânio Puerari for writing the original 2DFFT code on which the later versions of 2DFFT were based, and also William Ring for visually selecting spirals from the GOODS sample. This research has made use of the NASA/IPAC Extragalactic Database (NED) which is operated by the Jet Propulsion Laboratory, California Institute of Technology, under contract with the National Aeronautics and Space Administration. This research has made use of NASA's Astrophysics Data System. Parts of this research were conducted by the Australian Research Council Centre of Excellence for Gravitational Wave Discovery (OzGrav), through project number CE170100004.

**Conflicts of Interest:** The authors declare no conflict of interest. The funders had no role in the design of the study; in the collection, analyses, or interpretation of data; in the writing of the manuscript, or in the decision to publish the results.

## Appendix A. Spirality Code

The Spirality code package is written in MATLAB and can be found here: https://github.com/DeannaShields/Spirality (accessed on 1 September 2022).

*Appendix A.1. File: Spirality.m*

Spirality.m is the main workhorse of the Spirality code package. It computes the best-fit pitch angle for spirals in FITS format. Pitch angle coordinate systems, or templates, are computed with many different pitch angles, and the spiral is fit to each.

The output variables are as follows:

- **PITCHvsINNER** (Double)—A two-column array showing the galaxy's best-fit pitch angle, in degrees, as a function of inner measurement radius.

- **BESTFITPITCH** (Double)—The answer, i.e., the mean pitch angle in the PITCHvsIN-NER array, or the best-fit pitch angle of the galaxy.
- **ERR** (Double)—The total error in the BESTFITPITCH measurement. It is the standard deviation of the pitch angles in the PITCHvsINNER array, scaled by the range of visible spiral radii divided by the range of inner measurement radii, then added in quadrature with the input parameter InnerRadiusSpacing.

The inputs of Spirality are as follows:

- **FILE** (String)—The filename of the galaxy image. The file must be in *.FITS format, and the galaxy must be oriented face-on or deprojected to circular.
- **X0, Y0** (Positive doubles)—The center of the galaxy in Cartesian pixel coordinates.
- **VIS_INNER, VIS_OUTER** (Positive doubles, VIS_INNER < VIS_OUTER)—Visually estimated inner and outer radii, in pixels, of the galaxy's spirals. These inputs are used to compute the error bar, not to compute pitch angle itself.
- **MSMT_INNER1, MSMT_INNER2, MSMT_OUTER** (Positive doubles, MSMT_INNER1 < MSMT_INNER2 < MSMT_OUTER)—The code first measures the galaxy on an annulus with inner radius MSMT_INNER1, in pixels, and outer radius MSMT_OUTER, in pixels. It then repeats the process, increasing the inner radius incrementally to MSMT_INNER2. The outer radius is held constant at MSMT_OUTER. The best-fit pitch is the mean of pitch angles measured on all such annuli.
- **InnerRadiusSpacing** (Positive double)—Spacing, in pixels, between successive inner radii. As a starting point, we recommend measuring around 11 inner radii, meaning InnerRadiusSpacing $\sim$ (MSMT_INNER2—MSMT_INNER1)/10.
- **NAXIS** (Positive integer)—Number of spiral axes in each spiral template. We recommend between $2\pi R$ and $4\pi R$, where $R$ is the outer radius of the galaxy in pixels. Insufficient values of NAXIS will result in high-frequency, periodic variations in the fitting function, particularly in the loose end of pitch angle domain (that is, near $\pm 90°$).
- **MINP, MAXP** (Doubles, $-90 \leq$ MINP $\leq$ MAXP $\leq 0$ or $0 \leq$ MINP $\leq$ MAXP $\leq 90$)—Minimum and maximum pitch angles, respectively, in degrees, of the pitch angle measurement domain. Computation time diverges if zero is included in the domain.
- **PSTEP** (Positive double) Spacing, in degrees, between pitch angles of successive measurement templates. For coarse measurements, we recommend PSTEP = 1. For fine measurements, we recommend PSTEP = 0.2. Note that PSTEP is the minimum possible error in the pitch angle measurement.
- **AxisPointSpacing** (Positive double) The spacing, in pixels, between computation points on each spiral axis of each pitch angle template. As a starting point, we recommend AxisPointSpacing = 0.2. Computation time varies inversely with this quantity.
- **SMOOTH** (0 or 1)—A toggle for applying a 5-point moving average to the fitting function. If SMOOTH is 1, the moving average is applied; otherwise it is not. This feature is useful in smoothing high-frequency variations caused by an insufficient value of NAXIS. However, it can also affect the location of the peaks, so use with caution.
- **Save2D, Save3D** (0 or 1)—Toggles for saving the output files. If Save2D is 1, a 2-D graph of the fitting function vs. pitch angle will be generated for each inner radius. If Save3D is 1, a 3-D graph of the fitting function vs. pitch angle and inner radius will be generated. If either variable is set to 1, then a text file summarizing the results will be generated.

*Appendix A.2. File: SpiralArmCount.m*

The SpiralArmCount code counts the arms in a spiral image. It is intended to be used after the pitch angle has been determined. The user inputs the bulge or bar radius, the spiral radius, and the pitch angle. The code plots the median pixel value vs. phase angle for a spiral coordinate system with the given pitch angle. It then performs an FFT on that result, yielding the relative strength of each mode.

It should be noted that, like all FFT methods of spiral analysis, the mode is not actually the number of arms, but rather, a statement of symmetry. For example, an $m = 4$ galaxy has spiral arms that are $90°$ apart, even if there are fewer than four arms.

The file outputs of SpiralArmCount are as follows:

- Median pixel value vs. phase angle (.fig, .eps)—A graph showing one local max for each spiral arm.
- FFT (Counting function vs. Mode) (.fig, .eps)—A graph showing the strength of the counting function for each symmetric mode.

The passed variables of SpiralArmCount are as follows:

- M (Double array, $1 \times 10$)—The symmetry mode domain of the counting function.
- Power (Double array, $1 \times 10$)—The counting function.
- Count (Double)—The mode with the maximum value of the counting function, i.e., the number of spiral arms.

The inputs of SpiralArmCount are as follows:

- **FILE** (String)—The filename of the galaxy image. The file must be in *.FITS format, and the galaxy must be oriented face-on or deprojected to circular.
- **X0, Y0** (Positive doubles)—The center of the galaxy in Cartesian pixel coordinates.
- **INNER** (Positive double)—The inner radius, in pixels, of the spiral annulus. In other words, the radius of the bulge or bar.
- **OUTER** (Positive double)—The outer radius, in pixels, of the spiral annulus. In other words, radius of the galaxy.
- **PITCH** (Double, $-90 \leq \text{PITCH} < 0$, or $0 < \text{PITCH} \leq 90$)—The pitch angle, in degrees, of the galaxy. For *S*-windings, the pitch is positive. For *Z*-windings, the pitch is negative. Computation time diverges as PITCH approaches zero.

*Appendix A.3. File: GenSpiral.m*

The GenSpiral function produces an image file in FITS format that contains a synthetic spiral that is suitable for testing spiral analysis codes. The user specifies the number of arms, the central pitch angle, the amount by which the pitch angle varies with the radial coordinate, the spiral's radius in pixels, the thickness of the spiral arms, the amount of Gaussian noise, the luminosity gradient along the radial coordinate, and the radius of the bulge. Computation time for this code is negligible.

The file outputs are a FITS and a JPEG file containing the spiral image with the properties specified by the inputs. The passed variable is IMAGE, a two-dimensional array containing all the pixel values in the output image.

The inputs of GenSpiral are as follows:

- **M** (Integer)—The number of arms in the output spiral.
- **PCONST** (Double, $-90 \leq \text{PCONST} \leq 90$)—The pitch angle, in degrees, at the center of the spiral. If the spiral is logarithmic (specified by the input PSLOPE = 0), then PCONST is the pitch angle of the spiral.
- **PSLOPE** (Double)—Linear change in pitch angle, in degrees, from the spiral's center to the edge. In other words, the pitch angle at the center is PCONST, while the pitch angle at the edge is PCONST + PSLOPE.
- **RADIUS** (Positive integer)—Radius, in pixels, of the output spiral.
- **THICK** (Positive integer)—Thickness, in pixels, of the spiral arms.
- **INVSNR** (Positive double)—The reciprocal of the signal-to-noise ratio of the output image. If INVSNR = 0, no noise is added to the image. Otherwise, Gaussian noise is added such that the SNR is equal to 1/INVSNR.
- **GRADIENT** (0, 1, or 2)—Determines whether the spiral will have a galaxy-like luminosity profile or whether the spiral arms will have the same intensity throughout:
    - If GRADIENT = 0, then every pixel on the spiral has the same pixel value before the Gaussian noise is added.

- If GRADIENT = 1, then the spiral's luminosity is modeled after UGC 463, and the Gaussian noise has the same luminosity profile as the spiral.
- If GRADIENT = 2, then the spiral's luminosity is modeled after UGC 463, but the Gaussian noise distribution remains constant throughout the image.

- **FILESAVE** (0 or 1)—Toggle that determines whether the output files will be saved. If FILESAVE = 1, a FITS file and a JPEG file are output. If FILESAVE = 0, no files are output.
- **BULGERADIUS**—Radius, in pixels, of the circular bulge in the output spiral. If BULGERADIUS = 0, no bulge is added.

*Appendix A.4. File: SymPart.m*

This function expresses an image (matrix) as its *M*-fold symmetric part and its residual, where $M = 360°$ divided by the rotational symmetry angle. This code neither reads nor writes any file. Rather, the input and output images are both matrices of pixel values.

The passed (output) variables of this function are as follows:

- **SYM** (Double array)—The symmetric component of the input image.
- **RESID** (Double array)—The residual from the symmetric component of the input image. Adding RESID to SYM yields the original image array.

The inputs of this function are as follows:

- **IMAGE** (Double array)—The array containing the pixel values of the input image.
- **M** (Integer)—The harmonic mode to be extracted. For example, if M = 2, the code will compute the 2-arm (180°) symmetric component.
- **C0, R0** (Double)—The column and row, respectively, of the center of the spiral in the input image array. This is equivalent to the Cartesian X and Y in a FITS image.

*Appendix A.5. File: MultiSymPart.m*

This function expresses an image (matrix) as the sum of its 2- and 3-fold symmetric components, or else its 2-, 3-, and 4-fold symmetric components, and the residual. As with SymPart, the input and output images are both matrices of pixel values.

The inputs of this function are the same as for SymPart, but with one addition: HIGH_M gives the highest mode to be computed. HIGH_M is an integer with a value of either 3 or 4. We do not recommend using this code to compute modes higher than 4.

The passed (output) variable of this function are as follows:

- **MULTI_SYM** (Double array)—The sum of the specified symmetric components of the input image.
- **MULTI_RESID** (Double array)—The residual from the symmetric components. Adding MULTI_RESID to MULTI_SYM yields the original image array.

*Appendix A.6. File: Functions Called by Other Functions*

The following functions are not called by the user, but are necessary in order for the Spirality code to work properly. They should be downloaded into the MATLAB folder along with the user-called functions.

- Extract_Filename.m
- fitsread.m
- fitsheader.m
- fitswrite.m
- PeriodToDash.m
- RotateTheta.m

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
