# Peer review of "Spirality: A Novel Way to Measure Spiral Arm Pitch Angle"

_galaxies, doi:10.3390/galaxies10050100_

Round 1
Reviewer 1 Report
Dear Authors.
I spent a good time reading your manuscript. You software, Spirality seems a good way to determine arms of spiral galaxies.
In general I think that the manuscript is well written and the results are well presented.
As a reader I would like to have read something regarding alternatives other than logarithmic approaches, are there any? There are some papers regarding the fact that pitch angles are different for each arm for one single galaxy; there are other papers that suggest that pitch angles change with galactic distances; other suggest that corrotation can play an important role. None are discussed in you manuscript.
However, as a software approach, your results can be used by the astronomy community.
Finally, it will be great if you can include a table that make comparisons among your package and other similar and check results accuracy, computacional times, etc.
Are you planing to have a webpage where the community can upload an image and you return the number of arms and pitch angle?
Best regards.
Author Response
Thank you for your feedback. Although many galaxies do approximate logarithmic spirals, many do not. A non-logarithmic approach would therefore be good to look into for future work. As for galaxies with different arms of different pitches, our code can only spot that if the different pitches have different symmetries - for example, a two-arm spiral with a third “straggler” arm with a different pitch. We have included an example of such a galaxy in Figure 5. As for testing Spirality against a different code, we have tested it against a 2-dimensional Fast Fourier Transform (2DFFT) and summarized the results in Table 3. Your suggestion of converting using the algorithm in an interesting web page is an intriguing one; we will be considering it in the future.
Reviewer 2 Report
My paper is that this manuscript is not yet in a form where it is suitable for publication in a journal. It is intended as a publication to describe a new code Spirality, which is intended for public use. The manuscript needs to contain a step-by-step description of the algorithms in the code and how they operate.I found section 2.2 on the computational method unclear and difficult to follow. Similarly, Appendix A, which is meant to contain more details about the code does not contain any useful description of the algorithms.
Most of the paper contains code tests. The material is very poorly organized. It does not describe the motivation for the set of tests that are undertaken, nor discuss the results in a way that the reader can understand what the outcome was.
Author Response
Thank you for your feedback. For the sake of clarity, we have made extensive revisions to Section 2.2, where the algorithm is described step-by-step in words. Most notably, we now refer directly to the input and output variables. For example, the user determines the number of spiral axes in each pitch angle template using the input variable NAXIS. Therefore, we changed “The number of axes is chosen by the user” to “The number of axes is chosen by the user [input: NAXIS]”. While Section 2.2 describes the algorithm, Appendix A summarizes the input and output variables.
For the code tests, we revised the introduction to Section 3 to clarify the motivation for each family of tests: “In this section we present examples of pitch angle measurements on synthetically generated logarithmic spirals of known pitch, as well as on real galaxies. For synthetic spirals, the aim is to test the robustness of the algorithm by varying both the input parameters and the spiral properties. We then test the code's reliability on real galaxies by comparing its results to 2DFFT measurements of the same galaxies. Finally, we test the galaxy parameters under which the code is reliable by comparing measurements of high-resolution galaxy images to low-resolution images of the same galaxies.”
Reviewer 3 Report
This is the report of my review of manuscript galaxies-1892579 submitted by D. Shields and collaborators with the title "Spirality: A novel way to measure spiral arm pitch angle". The manuscript describes a software package that can be used to measure characteristics of the spiral structure of images without some of the assumptions usually adopted in previous work. The proposed method is interesting, useful, and its implementation is clearly described in the text. The manuscript is well written, although it sometimes deviates from what should be expected in a research article. For this reason, I think it needs some revision before I can recommend it for publication in Galaxies.
1) Sections of the manuscript read like a user manual and not so much as a research article. This is particularly clear in appendix A, where the authors list all the parameters admitted by each one of the files in the package, and in the frequent suggestions for use in the text. In my opinion, this is unnecessary in a research article and I suggest the authors to place it in a text to be distributed together with the package files.
The "Future work" section is also unusual in a research article... but I actually think it should be more frequently included.
2) In a similar fashion, I think the tests section is unnecessarily long. The examples in sec. 3.1 and 3.3 are enlightening, but the detail of tests presented in sec. 3.2 is will be useful for a small amount of readers only. Therefore, I suggest the authors to move sec. 3.2 to an appendix and discuss only a summary of those tests in the main text.
2.1) In contrast, the discussion in sec. 3.3.1 on the 4 galaxies for which Spirality and 2DFFT yield inconsistent results is lacking a discussion on the reason for the disagreement.
3) Maybe I missed it, but it would be useful to add a discussion on what to look for in galaxy images when deciding to use Spirality or 2DFFT.
4) I would definitively be among the first to answer the question of "Why measure pitch angles?" with a resounding "Because we can!" Yet, measuring pitch angles in a reliable and consistent way is useful for several reasons beyond the couple the authors mention in the sec. 1.2. The manuscript could be improved by extending the motivation behind their work, the context, and possible impact of the tool the authors developed.
Minor comments:
5) I would like to see references to the galaxy images used as examples in sec. 3.
6) The authors mention a couple of times the consumption of computer time their package requires. Have they considered implementing multi-threading and/or parallel processing? I am not very familiar with matlab, but I read that such capacity is available.
In a related note, why did they chose matlab? It seems to me that python is computationally more efficient and is more widely used in the astronomical community (and multi-threading is implemented in an almost trivial way).
This is more of a "reader" comment than a "reviewer" comment, and so I do not expect the authors to change the text if they do not consider it necessary.
Author Response
Thank you for your feedback.
Points 1) and 2): We believe our extensive testing is the backbone of the code’s reliability, and that the Appendices are helpful to anyone trying to use the code. We have published the tests (i.e., Section 3) and the Appendices in the Github repository, and may link them to ReadtheDocs.org as well.
Point 2.1): In Section 3.3.1, we have added a brief discussion of Spirality’s disagreement with 2DFFT for the galaxy UGC 1635, in addition to the discussions of the other three galaxies that were already in place. Here is the added discussion: “Although the error bars did not quite overlap, the Spirality's measurement was very nearly consistent with 2DFFT's. Because our error bars are computed from standard deviations, they are based on the assumption that the best-fit pitch follows Gaussian statistics. We would therefore expect slight disagreements among measurements a small fraction of the time.”
Point 3): We have added to the discussion of when to use Spirality vs. 2DFFD in the introduction to Section 4: “In general, we recommend Spirality when foreground stars cannot be subtracted, and especially when one spiral arm is brighter than the others. We recommend 2DFFT for very large image files, when computation time is an issue. Neither code is recommended for galaxies that are less than 30 pixels in radius.”
Point 4): Section 1 begins “The global, best-fit pitch angles (P) of galaxies are an ongoing topic of interest because they correlate to difficult-to-estimate properties such as bulge mass, central black hole mass, rotational velocity, and dark matter halo mass”. We have now added “Pitch angle is easier to measure than, and can be used as an estimator for, more fundamental properties.”
Point 5): We have included image source references are in Table 3.
Point 6): Implementing the algorithm in Python is a great idea!